# An Overview of the Effects of Lithium on Alzheimer’s Disease: A Historical Perspective

**DOI:** 10.3390/ph18040532

**Published:** 2025-04-05

**Authors:** Marcia Radanovic, Monique Patricio Singulani, Vanessa de Jesus R. De Paula, Leda Leme Talib, Orestes Vicente Forlenza

**Affiliations:** 1Laboratory of Neuroscience (LIM-27), Department and Institute of Psychiatry, Hospital das Clínicas da Faculdade de Medicina da Universidade de São Paulo (HCFMUSP), São Paulo 05403-010, SP, Brazil; marciaradanovic@gmail.com (M.R.); moniquesingulani@gmail.com (M.P.S.); vanessajrpaula@gmail.com (V.d.J.R.D.P.); ledatalib@gmail.com (L.L.T.); 2Instituto Nacional de Biomarcadores em Neuropsiquiatria (INBioN), Conselho Nacional de Desenvolvimento Científico e Tecnológico, São Paulo 05403-010, SP, Brazil; 3Centro de Neurociências Translacionais (CNT), Faculdade de Medicina da Universidade de São Paulo (FMUSP), São Paulo 05403-010, SP, Brazil

**Keywords:** lithium salts, lithium therapy, neurodegenerative diseases, Alzheimer’s disease, neuroprotection

## Abstract

Lithium was introduced into psychiatric practice in the late nineteenth century and has since become a standard treatment for severe psychiatric disorders, particularly those characterized by psychotic agitation. It remains the most effective agent for managing acute mania and preventing relapses in bipolar disorder. Despite potential adverse effects, lithium’s use should be carefully considered relative to other treatment options, as these alternatives may present distinct safety and tolerability profiles. The World Health Organization classifies lithium salts as ‘essential’ medications for inclusion in global healthcare systems. Over the past two decades, the growing recognition of lithium’s efficacy—extending beyond mood stabilization to include reducing suicide risk and inducing neuroprotection—has led to its incorporation into clinical practice guidelines. Current research, particularly from translational models, suggests that lithium’s pleiotropic effects benefit not only mental and brain health but also other organs and systems. This supports its potential as a therapeutic candidate for neurological conditions, particularly those associated with neurodegenerative processes. This article will discuss the historical background, discovery, and early experimentation of lithium in psychiatry. We will also review its mechanisms of action and discuss its potential in the treatment and prevention of neurodegenerative disorders, focusing on Alzheimer’s disease.

## 1. Introduction

Lithium is a soft, silvery-white alkali metal with an atomic mass of 6.941 u and an atomic number of 3 [1]. Its therapeutic potential has been recognized for millennia, with the first recorded use dating back over two thousand years. Ancient civilizations discovered the medicinal benefits of lithium-rich waters, which were believed to have healing properties for a range of physical and mental health conditions [2]. One of the earliest documented references occurs in the early second century C.E., when the Greek Soranus of Ephesus, a renowned physician expert of the time, recommended the use of alkaline waters to promote mental health. He recommended, particularly for individuals suffering from conditions such as mania and melancholia, to immerse themselves in these mineral-rich springs as a treatment form. Later, this practice led to the eventual identification of a link between lithium-containing mineral springs and the management of manic episodes. By the 19th century, the therapeutic use of water from various hydro-mineral resorts, including the famous Eau de Vichy, became widely popular. These mineral waters were believed to provide relief for a broad spectrum of both physical and mental health disorders, including mania, melancholia, epilepsy, arthritis, gout, and even cancer [2,3,4,5].

The widespread adoption of lithium marked the beginning of modern psychopharmacology and continues to be the most well-established mood stabilizer for bipolar disorder (BD) [6]. Moreover, substantial evidence demonstrates its efficacy in preventing suicide and treating neurodegenerative diseases, further solidifying its role as a potential therapeutic agent in psychiatry.

## 2. The Journey of Lithium: From Discovery to Clinical Applications

The discovery of lithium can be traced back to the mid-18th century and is credited to José Bonifácio de Andrada e Silva, a Brazilian statesman who played a key role in his country’s struggle for independence from Portugal. In addition to his political accomplishments, Andrada e Silva was also a scientist, and his notable contribution as a mineralogist was the identification of petalite crystals on the Swedish island of Utö, renowned for its rich iron mines, from which lithium was later extracted [7]. During this period, Reverend Edward Daniel Clarke, a professor of mineralogy at the University of Cambridge, was among the researchers who obtained access to petalite samples and published a detailed description of its constituents; however, a small fraction remained unidentified [2,5]. In 1817, while analyzing a petalite specimen in the laboratory of Jöns Jacob Berzelius, one of the founders of modern chemistry, the 25-year-old Swedish chemist Johan August Arfwedson identified a previously unknown element. Subsequently, Berzelius named the new element ‘lithium’, derived from the Greek word ‘lithos’ (stone), to reflect its mineral origin [2,4].

William Thomas Brande, an English chemist and geologist, became the first to isolate lithium in its pure form in 1821 by performing electrolysis on lithium oxide. It was not until 1855 that the British chemist Augustus Matthiessen and the German chemist Robert Bunsen developed a process that enabled the large-scale production of lithium through the electrolysis of lithium chloride. This discovery not only facilitated the systematic investigation of lithium’s properties across various scientific and industrial domains but also marked a pivotal turning point in the initiation of its widespread commercialization for various applications. Furthermore, with the availability of sufficient quantities, it was incorporated into medical treatments [2,4,5].

The early medical use of lithium began with its administration for the treatment of gout, and later expanded to include conditions related to the bladder, kidneys, and gallstones. In 1843, Alexander Ure, a Scottish surgeon, proposed using lithium carbonate to treat renal stones, based on his observation that it could dissolve uric acid crystals. In 1860, Ure suggested treating patients with urinary stones by injecting lithium salts into their urinary bladders, but the treatment was not clinically successful. His findings, on the other hand, contributed to the belief that lithium could also be beneficial in the treatment of gout, thus enabling continued research on its effects [2,5,8].

Later, in 1847, Alfred Baring Garrod, a London physician, detected elevated concentrations of uric acid in the blood of patients suffering from gout. He began investigating the use of lithium salts as a treatment for dissolving uric acid crystals. His research culminated in the publication of ‘The Nature and Treatment of Gout and Rheumatic Gout’ in 1859, a work that significantly contributed to the broader understanding and application of lithium in medical treatments [9]. It was also noted in that same decade that a significant number of individuals diagnosed with gout exhibited symptoms of depression. Garrod hypothesized that the effects of excessive uric acid could impact the central nervous system, resulting in what he termed ‘brain gout’, a condition he associated with depression, based on his empirical observations. He later recommended the use of lithium to treat mental illness [9]. However, it was the writings of Alexander Haig (1853–1924) on ‘uric acid diathesis’ that ultimately contributed to the widespread use of lithium. Haig, in agreement with Garrod’s hypothesis, suggested that excess uric acid was responsible for a wide range of ailments, including neuropsychiatric disorders, and recommended the use of lithium salts to treat these conditions [10].

The use of lithium in psychiatry dates back to the 1870s, when American surgeon general William Hammond, a professor at the Bellevue Hospital Medical College, presented evidence of effectively treating acute manic patients with lithium bromide. However, he was unable to determine if the outcomes were attributable to lithium or bromide [11]. In the 1880s and 1890s, Danish psychiatrists Carl Georg Lange and Frederick Lange were the first to systematically use lithium for both acute and prophylactic treatment of melancholic depression. The Lange brothers reported the successful treatment of approximately 2000 depressed outpatients with lithium carbonate over a 20-year period [12]. Carl also noted that some patients with depression treated with lithium carbonate showed elevated levels of urinary sediment, which he attributed to the liberation of uric acid. Even though the uric acid hypothesis was eventually refuted, lithium treatment provided tangible benefits in various general medical and neuropsychiatric conditions [12]. However, ongoing debates regarding the validity of the hypothesis in relation to neuropsychiatric disorders and reports of severe intoxications and fatalities contributed to the gradual decline in the use of lithium [13].

## 3. The Modern Revival of Lithium

Interest in lithium reemerged during the early 19th century when David Culbreth, a pharmacology professor at the University of Maryland, described lithium bromide as ‘the most hypnotic of all bromides’ and advocated its use in the treatment of epilepsy. Despite this, lithium salts were not integrated into the treatment of mental illnesses until the late 1940s [14].

The foundational research that contributed to the advancement and dissemination of the use of lithium salts as a successful treatment for manic-depressive psychoses and BD is credited to Australian psychiatrist John F. J. Cade. Based on his interventional studies, Cade demonstrated the antimanic effects of lithium, describing it as ‘lithium salts might well be an important addition to the anti-convulsant armamentarium’, and detailing his investigation of the ‘protective effect’ of lithium [15]. His seminal paper was published in the 3 September 1949 issue of the *Medical Journal of Australia*, under the title ‘Lithium Salts in the Treatment of Psychotic Excitement’ [16]. Cade hypothesized that the etiology of ‘manic-depressive insanity’ was due to a dysregulation of the body’s metabolism. He proposed that mania resulted from a ‘state of intoxication of a normal product of the body circulating in excess’, whereas melancholia was a ‘corresponding deprivative condition’. To investigate these metabolites—the presumed ‘toxic agent’—Cade conducted what he described as an ‘extraordinarily crude differential toxicity test’ to identify substances excreted in excess in the urine of patients with psychiatric disorders, which could contribute to distinguishing between those with mania and those with depression based on their symptoms [16].

Cade used a scientific approach that involved injecting samples of concentrated urine from manic, melancholic, and schizophrenic patients, as well as from healthy people, into the peritoneal cavity of guinea pigs. Although the mode of death in the animals continued to be consistent, Cade’s experiment indicated that the urine of human subjects diagnosed with mania was harmful to animals compared to urine from other individuals. Initially, Cade hypothesized that urea might be the ‘toxic agent’, as intraperitoneal injections of this urine constituent caused the same mode of death in the guinea pigs. However, during his analysis, Cade noted that, despite similar urea levels, the urine of manic patients was significantly more toxic than that of non-manic individuals. He suggested that another substance might be contributing to the increased toxicity of urea after this finding. At that time, Cade had already noticed that uric acid seemed to increase the toxicity of urea. However, due to the water-insoluble nature of uric acid, lithium urate, the most soluble urate, was used as an alternative. Cade observed that the toxicity was significantly lower than expected, a phenomenon he described as the ‘great paradox’. He speculated that the lithium-ion might be exerting a protective effect. To further explore the findings of his previous experiment, Cade used a urea solution with lithium carbonate instead of lithium urate. He observed a reduction in lethality in the guinea pigs, which confirmed that lithium itself had a protective effect against the toxic impact of urea [16]. Cade also observed that when lithium salts were tested alone (i.e., without urea), they caused anxiolytic effects in guinea pigs [15,16].

Cade’s unexpected finding led him to shift his focus toward investigating the tranquilizing effects of lithium on his patients. To establish a safe dosage of lithium salts for human use, he first ingested them himself. Following this, he conducted a pilot trial involving patients diagnosed with mania, melancholia, and early dementia, which led to the hypothesis that lithium-ion deficiency might contribute to these conditions [17,18]. Cade treated 10 manic patients, 6 schizophrenic patients, and 3 melancholic patients with lithium, specifically lithium citrate and lithium carbonate. All patients exhibited recovery within a few days to weeks; however, relapse was observed upon discontinuation of lithium or in cases of non-compliance. While the psychotic symptoms of schizophrenic patients remained unchanged, a notable reduction in restlessness and agitation was observed, with three patients becoming calm and responsive for the first time in years. However, despite lithium treatment, melancholic patients continued to experience distress [17,18].

Although Cade’s interventional studies lacked standardized measurement methods, statistical rigor, and other methodological elements characteristic of contemporary research, his findings significantly influenced psychiatric practice [17,18]. Despite these promising results, the research faced challenges due to lithium’s side effects, which, in some cases, led to toxicity and even fatalities. Two years later, Cade’s first patient (Case I, W.B.) died of lithium poisoning due to the prescribed doses, prompting him to discontinue his research and cease prescribing lithium [13,15,17].

## 4. The Legacy of Lithium: From Therapeutic Validation to Approval

Although lithium treatment was considered controversial, primarily due to the occasional occurrence of severe toxic complications, the research on lithium remained ongoing throughout the early 1950s [2,5].

The first clinical report in the global literature following Cade’s publication appeared in 1951 in the *Medical Journal of Australia*. The study, entitled ‘The Lithium Treatment of Maniacal Psychosis’, was accomplished by Edward M. Trautner at the Department of Physiology at the University of Melbourne and Charles H. Noack at Mont Park Mental Hospital, Melbourne [19]. They conducted an open trial involving over 100 psychiatric patients with various diagnoses, who were treated with 0.6 g of lithium carbonate or 1.2 g of lithium citrate for durations ranging from two weeks to over one year. As a result, in most cases, Trautner et al. reported that ‘lithium treatment was found to be very beneficial in terminating and preventing the maniacal phase in cases of mania, hypomania and recurrent mania’, and added that ‘no cumulative effects or incidents of grave intoxication were observed in patients’. A significant aspect of this study was the introduction of blood lithium level assessment in all patients through flame spectrophotometry assays. With its approach, Trautner emphasized the importance of determining lithium levels for both safety monitoring and the validation of optimal therapeutic plasma concentrations [19].

In subsequent years, Trautner’s findings were confirmed by other studies; however, considerable skepticism remained regarding the long-term use of lithium, primarily due to concerns about the relationship between appropriate dosage, plasma lithium levels, and toxicity, which were widely recognized as unresolved issues at the time [20]. In 1995, Trautner presented notable findings on the pharmacokinetic properties of lithium, focusing on its effects on ionic balance, retention, excretion, and toxicity. These observations were performed on patients receiving lithium salts during the manic phase compared with their return to euthymia, as well as on healthy individuals for comparison. In his study published in *The Medical Journal of Australia* and titled ‘The Excretion and Retention of Ingested Lithium and Its Effect on the Ionic Balance of Man’, Trautner et al. reported that 90% to 95% of ingested lithium was excreted in urine, with most eliminated within 6–8 h and the rest over 2 weeks [21]. Lithium ingestion caused disturbances in water and ionic balance, particularly sodium (Na^+^), for up to 2 days. With prolonged moderate intake, a stable state was reached where lithium excretion matched intake, and ionic imbalances decreased. In this stable state, lithium retention equaled one and a half daily doses, and plasma levels correlated with the dose. However, in patients on high doses, stability was not maintained, and clinical instability, such as low plasma Na^+^ levels and susceptibility to toxic complications, was observed. Trautner also identified early symptoms of lithium poisoning, with a danger level at 3 mEq/L of plasma and a lethal level at 4–5 mEq/L [21].

The understanding of lithium’s therapeutic effects further progressed through investigations by Danish psychiatrist Mogens Schou and other researchers, who conducted a series of studies under more rigorous conditions. This led to a randomized controlled clinical trial on lithium treatment in mania patients, published in 1954, which demonstrated its efficacy for the majority of individuals with BD [22]. In 1967, Danish psychiatrist Poul C. Baastrup, along with Schou, demonstrated that long-term lithium treatment could not only alleviate manic episodes but also prevent depressive recurrences [23]. Confirming these findings, Baastrup and his collaborators published a 1970 study in *The Lancet* titled ‘Prophylactic Lithium: Double-Blind Discontinuation in Manic-Depressive and Recurrent-Depressive Disorders’. This double-blind discontinuation clinical trial included fifty patients with manic-depressive illness and thirty-four with recurrent endogenous depression, all of whom had been receiving open lithium treatment. The results demonstrated that lithium carbonate exerted a prophylactic effect in preventing further recurrences of endogenous affective disorders [24], thereby reinforcing its therapeutic efficacy and prophylactic potential [25].

While lithium gained recognition in several countries during the 1950s [19,22,26], interest in its use in the United States did not grow until a decade later. In 1960, Australian psychiatrist Sam Gershon, an early pioneer of lithium research in Melbourne, collaborated with Trautner to conduct a major clinical trial in the United States during the 1960s [27]. This study represents the first report of lithium treatment in North America, with seventeen patients diagnosed with manic excitement being treated at the Royal Victoria Hospital in Montreal. As a result, all patients, except for one, responded positively to lithium treatment. The authors also showed that lithium dosages ranging from 1.2 to 5.4 g, for which no poisoning occurred, resulted in maximal plasma levels of 0.8–2.8 mEq/L and highlighted lithium’s specificity for the treatment of manic excitement [27,28]. They also concluded that there were no significant barriers to the widespread use of lithium, which is one of the few available targeted therapies in modern psychiatry [29].

In fact, lithium was banned from medical practice in the US until the 1970s because of its unwanted effects. However, in 1968, the clinical significance of lithium was recognized in a special section of the *American Journal of Psychiatry*. In the early 1970s, the U.S. Food and Drug Administration (FDA) approved the use of lithium for the treatment of mania, and in 1974, it was approved for maintenance therapy of patients with mania, making the United States the 50th country to introduce it to the market [5,29]. Lithium had also been approved for medical use in several other countries, including France in 1961, the United Kingdom in 1966, Germany in 1967, and Italy in 1970 [4]. In retrospect, as evidence of lithium’s usefulness increased, strong support from the pharmaceutical industry might have been anticipated. However, since lithium is a naturally occurring element, making it an inexpensive, non-patentable drug, no American pharmaceutical company was willing to manufacture it, even after FDA approval, due to its lack of profitability. As a result, lithium research in the late 19th century progressed slowly, particularly in determining its safety limits for clinical use [3,6,28].

## 5. Broadening the Use of Lithium in Neurocognitive Functions

Since its FDA approval, further research has substantiated lithium’s therapeutic and prophylactic efficacy in managing acute mania and preventing bipolar disorder, while its application has extended to other medical fields. However, long-term administration has been linked to adverse effects and complications, including memory impairment and kidney dysfunction. This caught the attention of the medical community, raising concerns about the potential for lithium to induce irreversible memory disorders in patients with psychiatric illnesses [30]. However, a double-blind crossover clinical trial conducted in patients with affective disorders demonstrated that lithium carbonate treatment (0.5–2.2 mEq/L) led to a slowdown in performance on certain perceptual-motor tests but did not impair memory function [31]. In contrast, another study found that bipolar patients treated with lithium carbonate recalled significantly fewer words compared to unmedicated patients, suggesting a potential impact on verbal memory [32]. Further comparisons with patients diagnosed with affective disorders and treated with tricyclic antidepressants also indicated that the lithium-treated groups exhibited memory impairment [33]. Presumably, the adverse effects of lithium on cognition are associated with the use of higher doses. Furthermore, it is important to determine whether there is conclusive evidence of such effects, considering the psychiatric status of lithium-treated patients and whether the battery of memory and cognitive tests used was the most appropriate for each case [34,35,36].

The positive effects of lithium treatment on neurobiological and neurocognitive functions began to be demonstrated in the late 1970s. Physicians Kenneth H. Williams and Gerald Goldstein were the first to suggest, in a paper published in the *American Journal of Psychiatry* in 1979, that lithium could play a role in treating dementia-like symptoms. Their study, titled ‘Cognitive and Affective Responses to Lithium in Patients with Organic Brain Syndrome’, examined patients with a combination of dementia and agitated depression who were hospitalized for organic brain syndrome. These patients received lithium carbonate at doses of 900–1200 mg/day. Of the 10 cases they evaluated, 8 were described as having shown ‘improved dramatically in affect and cognition with lithium after having shown no significant change with other psychotropic medications’ [37]. In 1990, Calil et al. [38] showed that healthy individuals taking lithium, with serum levels ranging from 0.6 to 1.0 mEq/L, exhibited slower reaction times but demonstrated improved performance on cognitive tasks. In the same decade, Clarke et al. [39] studied the effect of lithium on cerebrospinal fluid (CSF) amyloid-beta (Aβ) precursor protein (APP)—a transmembrane protein that plays a central role in the pathophysiology of Alzheimer’s disease (AD)—in dementia patients. Their findings showed that the mean APP695 value, the most common APP isoform in the brain and a key player in Aβ production—a hallmark of AD, was lower in demented patients who had received lithium salts compared to those treated with other medications, such as antidepressants, antipsychotics, and tranquilizers. This study sparked interest in lithium’s neuroprotective properties for AD treatment, as lithium appeared to selectively modulate APP, may influence the disease’s pathological processes. But the interest in lithium’s therapeutic action grew stronger when Paul Grof (2010) introduced the concept of ‘excellent lithium responders’, a specific subgroup of BD patients in whom lithium monotherapy can restore normality in clinical profile and cognitive function, compared to age-matched, healthy control subjects [40]. A few years later, considerable evidence accumulated showing the neuroprotective effect of lithium at therapeutic and subtherapeutic doses [6,36,41,42,43].

## 6. Lithium and Its Impact on Human Health

Natural lithium is present in human nutrition through food intake, predominantly in plant-based foods and drinking water. Traces of lithium have been detected in the human body, with an average concentration of approximately 7 mg [44]. Lithium is an essential trace element involved in various cellular processes, influencing the functions of multiple enzymes, hormones, and vitamins. The fact that lithium is a monovalent cation with chemical properties similar to potassium (K^+^) and Na^+^ suggests that it plays a key role in maintaining brain homeostasis. Lithium is equally distributed between intra- and extracellular compartments and enters cells mainly through voltage-gated Na^+^ channels, with its influx exceeding its efflux. Since lithium enters cells through electrical signaling, it accumulates more selectively in neurons with higher synaptic activity. As a result, lithium is more likely to interact with neurotransmitters and receptors in the brain, modulating neuronal plasticity, membrane components, transport mechanisms, and intracellular signaling molecules. At a structural level, it is via such a mechanism that lithium appears to preserve or increase the volume of brain structures involved in emotional regulation, such as the prefrontal cortex, hippocampus, and amygdala, possibly reflecting its neuroprotective effects [45,46]. This is supported by evidence of post-mortem studies that have indicated that the cerebellum, cerebrum, and kidneys retain more lithium than other organs, suggesting its potential influence on signaling pathways in key brain regions [1,44,45,47].

In addition to lithium, the other alkali metals, including Na^+^, K^+^, rubidium (Rb), cesium (Cs), and francium (Fr), have some function in the organism. In the human body, only Na^+^ and K^+^ are essential and vital to life, playing crucial roles in various physiological processes, such as the conduction of electrical impulses in excitable cells. Even a slight variation in their concentrations, either in the intra- or extracellular compartments, can lead to acute dysfunction of vital organs such as the brain, heart, and kidneys, potentially resulting in a life-threatening state. In contrast, Rb and Cs are considered non-essential elements for humans and have no known biological role [45,48].

According to nutritional studies, dietary lithium intake is necessary, provided it does not exceed 1 mg/day for a 70 kg adult (14.3 µg/kg body weight) [44,47]. Low-level exposure to naturally occurring lithium in the environment, such as in drinking water and certain food, including vegetables, fruits, and grains, e.g., cauliflower, tea plants, coriander leaves, tomatoes, garlic, nutmeg, cumin seeds, onions, green chilies, rice, and wheat, has been shown to benefit mental health [44]. Although natural lithium intake is significantly lower than therapeutic doses, studies have documented that low concentrations in drinking water consumed daily are associated with increased human lifespan. This effect is attributed to lithium’s anti-suicidal properties, its ability to prevent depressive states, and its role in reducing the incidence of dementia; however, the underlying neurobiological mechanisms are not yet fully understood [3,36,49].

Another important characteristic of lithium is its ability to easily distribute throughout the body via both intra- and extracellular fluids, due to its free protein binding. Lithium can cross the placenta to reach the developing fetus, and is also secreted in breast milk. As a result, lithium concentrations in both mothers and fetuses attain equilibrium, suggesting that it may contribute to neurodevelopmental health. In addition, during breastfeeding, the lithium concentration in milk decreases to approximately half of the maternal serum levels [46,50]. The control of lithium levels is particularly challenging during pregnancy due to normal physiological changes, especially in renal function. Increased glomerular filtration rate during pregnancy leads to decreased blood lithium concentrations and an increased risk of relapse in women with mania. As a result, clinicians often increase the lithium dose during pregnancy. However, later in pregnancy and during the early postpartum period, when glomerular filtration rate returns to preconception levels, the increased dosage may lead to toxic blood lithium levels [50]. Wesseloo et al. [50] showed that lithium initiated during pregnancy, either as treatment for an episode or as an alternative to a mood stabilizer, reduced the risk of teratogenicity when started before conception and continued throughout pregnancy and the postpartum period. The lowest blood lithium levels occurred during the second trimester; in the third trimester and postpartum period, lithium levels progressively returned to preconception levels, highlighting the importance of controlling lithium during this critical gestational period [50].

## 7. Managing Lithium: Therapeutic Index and Risk to Toxicity

Lithium was the first specific psychotropic agent, with sedatives such as bromides and paraldehyde, being its predecessors. Currently, it remains the benchmark for the treatment of bipolar disorder and is considered the standard reference among mood stabilizers. Although there are other approved medications for this condition, the use of lithium must be carefully assessed in comparison to alternative treatments, given that these alternatives may have differing safety and tolerability profiles. Nonetheless, the management of lithium treatment is challenging due to its narrow therapeutic index and the impact on kidney function, both of which heighten the risk of toxicity [51].

Meticulous attention to dosing, monitoring, and titration is essential. Lithium is prescribed for managing acute manic and mixed episodes, as well as for long-term maintenance treatment in patients aged 12 or older. The therapeutic dose typically ranges from 300 to 2700 mg/day, with the optimal steady-state serum concentration of lithium falling between 0.7 and 1.2 mEq/L. In elderly individuals, lower doses of lithium are required, with a mean dose just above 400 mg/day, to achieve the desired serum concentration of approximately 0.5 mmol/L. For children over the age of 12, serum lithium concentrations should generally align with those observed in younger adults, ranging between 0.6 and 1.2 mmol/L. Serum concentrations of lithium exceeding 1.5 mEq/L may lead to symptoms of mild intoxication, including fatigue, tremor, nausea, diarrhea, blurred vision, vertigo, confusion, and increased deep tendon reflexes. Levels surpassing 2.5 mEq/L may cause more severe neurological complications, such as seizures, coma, cardiac dysrhythmia, and permanent neurological impairment, often affecting the cerebellum. Careful management is essential when using lithium, and it should be avoided in patients with a history of acute myocardial infarction, acute kidney failure, or certain rare heart rhythm disorders [51,52].

## 8. Advancing Lithium Research Through Translational Studies

Despite its extensive clinical use for over eight decades, uncertainties remain regarding the mechanisms underlying lithium’s effects on mood and behavior. Lithium is often labeled a ‘dirty’ drug due to its interaction with various intracellular signaling pathways, including Wnt/β-catenin, adenylate cyclase, telomerase, bisphosphate nucleotidase, β-arrestin, and cyclooxygenase. Additionally, it affects the actions of GABA (gamma-aminobutyric acid) and NMDA (*N*-methyl-D-aspartate) receptors, and contributes to calcium homeostasis [53,54].

The primary molecular targets of lithium are the enzymes inositol monophosphatase (IMPase) and glycogen synthase kinase-3 beta (GSK-3β). These enzymes play critical roles in various cell types, directly or indirectly influencing metabolic processes such as apoptosis, autophagy, cytoskeletal remodeling, gene regulation, the cell cycle, neurotrophic support, energy metabolism, mitochondrial function, oxidative stress, and the inflammatory response [49].

Lithium directly inhibits IMPase activity by noncompetitively displacing Mg^2+^ from the enzyme’s catalytic sites, reducing inositol triphosphate formation [55]. This, in turn, modulates several intracellular pathways relevant to neuropsychiatric disorders, particularly by stimulating autophagy [56]. The autophagic effect of lithium may facilitate the clearance of aggregation-prone proteins, such as Aβ, phospho-tau (p-tau), and damaged mitochondria, potentially influencing the pathogenesis of AD [57,58].

GSK-3β is a constitutively active enzyme involved in the organization and remodeling of the cytoskeleton [59,60]. Lithium inhibits GSK-3β activity through two primary mechanisms: (a) directly, by competing with Mg^2+^ ions for binding to the enzyme’s catalytic site, and (b) indirectly, by inducing phosphorylation of the serine-9 residue of GSK-3β, leading to conformational changes that inactivate the enzyme [53,54,58]. These indirect mechanisms activate intracellular kinases, such as Akt, while inhibiting the intracellular protein phosphatase 2A (PP2A) [60]. GSK-3β inhibition also interferes with the Wnt/β-catenin signaling pathway, leading to the intracellular accumulation of β-catenin, which facilitates its entry into the nucleus to regulate the transcription of target genes and modulates several downstream pathological processes [60,61,62]. Ultimately, this reduces Aβ deposition, enhances neurogenesis, and improves mitochondrial bioenergetics [53,63,64,65]. Additionally, lithium can inhibit the mRNA transcription of GSK-3β, further reducing the enzyme’s availability [58,66].

Neuroinflammation plays a crucial role in neurodegenerative diseases. The overactivation of microglia and astrocytes, along with the inflammatory molecules they produce, can disrupt the neuronal microenvironment and lead to cognitive impairment [67]. When stimulated by immune responses or tissue damage, reactive pro-inflammatory microglia, activated by interferon-γ and tumor necrosis factor-α (TNF-α), release pro-inflammatory cytokines such as interleukin-1β (IL-1β), IL-6, IL-18, and TNF-α, along with nitric oxide (NO) and reactive oxygen species (ROS). These substances are associated with increased neurotoxicity [67,68]. Lithium treatment has the potential to reduce the production of pro-inflammatory factors, including IL-1β and TNF-α, as well as other neuroinflammatory biomarkers in animal models of AD [41,42,49,63,69].

## 9. Exploring Lithium’s Neuroprotective Potential

Evidence of lithium’s neuroprotective effects comes from experimental models and neuroimaging studies. Chronic lithium use in patients with BD is associated with an increase in the volume of cerebral gray matter and improved tissue viability [70,71,72]. Furthermore, these patients show a decreased prevalence of dementia compared to patients treated with other mood stabilizers. This benefit appears to be independent of therapeutic response parameters, suggesting that it may be due to lithium’s effects on brain systems responsible for maintaining homeostasis, including second messenger pathways, cellular communication, and cell viability [73].

Lithium-mediated inhibition of GSK-3β is causally linked to its effects on key pathogenic pathways of AD, particularly the Aβ cascade and hyperphosphorylated tau (p-tau). The accumulation of Aβ and p-tau is responsible for the formation of amyloid plaques and neurofibrillary tangles (NFTs), which are the key pathological hallmarks of AD. In the Aβ cascade, the APP can be cleaved by α-secretases, which generate non-toxic APP fragments, or by β- and γ-secretases, which produce toxic peptides through the amyloidogenic pathway. As previously mentioned, elevated levels of nuclear β-catenin, caused by GSK-3β inhibition, result in its interaction with transcription factor 4 (TCF4). This interaction inhibits the transcription of the main enzyme related to APP cleavage at the β1 site (BACE1), ultimately reducing Aβ production [49,60]. Additionally, lithium attenuates the activity of γ-secretase and enhances the clearance of Aβ by improving the function of the blood–brain barrier (BBB) microvessels, promoting Aβ efflux through cerebrospinal fluid (CSF) bulk flow [74].

The deposition of hyperphosphorylated tau protein (p-tau) leads to the formation of intraneuronal paired helical filaments and NFTs. This process disrupts the microtubule structure, impairs normal axonal transport of nutrients and signaling, and ultimately results in cell death. Lithium may help prevent the progression of tau neuropathology in AD by inhibiting GSK-3β and modifying the expression or activity of other tau kinases, such as protein kinase A (PKA), Akt (PKB), and calcium/calmodulin-dependent kinase II (CaMKII) [60,66,75]. Furthermore, lithium increases the expression of B-cell leukemia 2 family protein (Bcl-2), a cytoprotective protein that promotes axon regeneration by inhibiting apoptosis and supporting the remodeling of the neuronal cytoskeleton [76]. In experimental models, lithium has been shown to stimulate neurogenesis in the dentate gyrus of the hippocampus. Among lithium’s neuroprotective effects, the stimulation of the synthesis and release of neurotrophins, such as brain-derived neurotrophic factor (BDNF) and vascular endothelial growth factor (VEGF), is also noteworthy [42,49,58]. Figure 1 illustrates the mechanism through which lithium exerts neuroprotection against the neuropathogenic processes of AD.

In summary, the neuroprotective actions of lithium against neuropathogenic mechanisms of AD and other neurodegenerative disorders are partly due to its intrinsic effects on oxidative stress, autophagy, apoptosis, mitochondrial function, and neuroinflammation. Evidence supporting this hypothesis has emerged from the outcomes of translational research [73]. Controlled studies demonstrated that prolonged use of lithium carbonate in subtherapeutic doses could slow cognitive and functional decline while reducing the transition to dementia in a population. Notably, this clinical effect was accompanied by changes in the profile of AD CSF biomarkers [77,78,79].

Lithium is being explored as a potential disease-modifying agent/treatment not only for BD and AD but also for other neurodegenerative disorders such as Parkinson’s disease [41], Huntington’s disease [80], amyotrophic lateral sclerosis [81], and stroke [42]. The regulatory effects of lithium can vary based on factors such as the target tissue, duration of exposure, and tissue concentration at the site of action. Consequently, some biological effects may occur at lower tissue concentrations than those typically encountered in the treatment of mood disorders. Ecological studies conducted globally have shown an inverse relationship between suicide rates and lithium concentrations in groundwater, drinking water, and clinical settings [82,83]. Similar associations have been observed between environmental lithium and psychiatric hospitalization rates, incidents of violence [84,85] and the prevalence of dementia [36]. These findings suggest that the intake of lithium in low doses may result in long-term beneficial outcomes [43,77,78,83,86,87,88,89].

## 10. Conclusions

The evolution of lithium therapy highlights the progression of psychiatric medicine, from initial skepticism to its establishment as an essential therapeutic tool, underscoring the importance of continued research and clinical advancements in mental health treatment. Research from both preclinical and clinical models supports the idea that lithium has neurotrophic and neuroprotective effects. These beneficial effects may be due to its effects on various cellular metabolic processes crucial for neuronal survival, neuroplasticity, transcriptional regulation, energy metabolism, and protection against neurotoxic damage. Some of these mechanisms are directly associated with the central pathogenic processes of specific diseases, while others may reflect general responses that enhance intrinsic neuronal resilience and foster neuroprotective effects.

It is important to note that the clinical use of lithium should be limited to conditions where its efficacy has been well established by scientific evidence, such as mood disorders. For other health conditions, further research is needed to support new therapeutic indications. Lithium treatments, including off-label use, should consider factors such as dosage, the specific health condition, age, and potential drug interactions.

In light of this, it is evident that future research should aim to further explore the biochemical mechanisms of lithium. Investigating the molecular pathways it influences will give us insight into its clinical efficacy and interaction with cellular processes. This enhanced knowledge will improve lithium’s current use in mood disorders and possibly reveal new therapeutic targets. A more profound understanding of lithium’s mechanisms will help develop more precise treatment strategies and novel pharmacological approaches.

## Figures and Tables

**Figure 1 pharmaceuticals-18-00532-f001:**
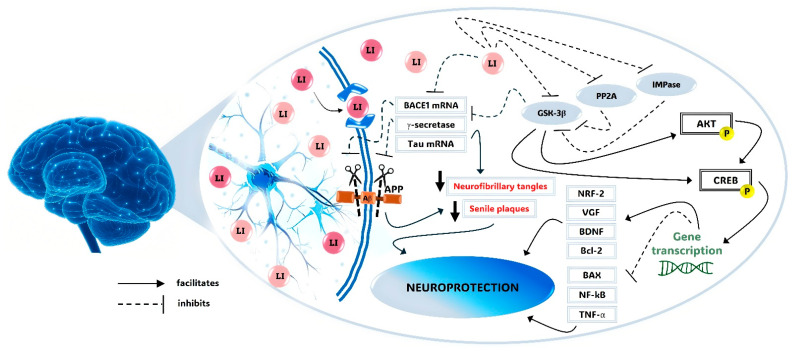
Molecular mechanisms of lithium in Alzheimer’s Disease. Alzheimer’s disease (AD) is characterized by the accumulation of senile plaques and neurofibrillary tangles (NFTs) in the brain, disrupting normal neuronal function. Lithium modulates the production and clearance of amyloid-beta (Aβ) by inhibiting β-site APP cleaving enzyme 1 (BACE1) mRNA expression and decreasing γ-secretase activity, thereby reducing amyloid precursor protein (APP) cleavage and Aβ production. Additionally, lithium reduces NFT formation by lowering tau mRNA expression and inhibiting glycogen synthase kinase-3 beta (GSK-3β), which in turn decreases tau phosphorylation levels and results in neuroprotection. At the same time, through the inhibition of GSK-3β, lithium induces the activation of protein kinase B (Akt) and cyclic adenosine monophosphate (cAMP) response element-binding protein (CREB) signaling pathways, thereby upregulating the mRNA expression and protein levels of nuclear factor erythroid 2-related factor 2 (NRF-2), neurosecretory protein VGF, brain-derived neurotrophic factor (BDNF), B-cell lymphoma 2 (Bcl-2), and other neuroprotective molecules. Conversely, by this same pathway, lithium downregulates the mRNA expression and protein levels of Bcl-2–associated X protein (BAX), nuclear factor kappa-light-chain-enhancer of activated B cells (NF-κB), tumor necrosis factor-alpha (TNF-α), and other molecules involved in neuroinflammation and apoptosis. Furthermore, by inhibiting protein phosphatase 2 (PP2A) and inositol monophosphatase (IMPase), lithium reinforces GSK-3β inactivation, promoting neuroprotective responses.

## Data Availability

No new data were created or analyzed in this study. Data sharing is not applicable to this article.

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
