# Peer review of "An Overview of the Effects of Lithium on Alzheimer’s Disease: A Historical Perspective"

_pharmaceuticals, 2025, doi:10.3390/ph18040532_

Round 1

Reviewer 1 Report

Comments and Suggestions for Authors

See attachment for suggestions/recommendations to the authors

Comments on the Quality of English Language

See attachment

Author Response

Responses to Reviewer #1

The review summarized the current advances, understanding, and the potential role of lithium as “a disease modifying agent” for the treatment of Alzheimer’s disease (AD) and other neurodegenerative diseases. The review is concise and informative and will be a good reference for both preclinical and clinical investigators who are focused on the current advances in AD and other neurodegenerative disorders. Although concisely written, the use of more technical and scientific terms will improve the impact and readership of this review. In this regards, please find below some minor suggestions to this effect.

Answer: We sincerely appreciate your invaluable feedback. We have carefully addressed each point raised and made the necessary revisions to the manuscript. For each comment, we provide the corresponding page and line numbers for the respective corrections.

Minor concerns:

  1. In the abstract, second to the last sentence (line 27), I suggest paraphrasing to “we will discuss the historical aspects, discovery, and early experimentation”

Answer: We appreciate you bringing this to our attention. The necessary corrections have been made in the revised manuscript and can be found on Page 1, lines 28-29.

“This article will discuss the historical background, discovery, and early experimentation of lithium in psychiatry.”

  1. In abstract, last sentence (line 28), I suggest paraphrasing to “”and discuss its potentials in the treatment and prevention of”

Answer: Thank you for your valuable feedback. The revised manuscript has been updated in accordance with your suggestions (Page 1, lines 29-30).

“We will also review its mechanisms of action and discuss its potentials in the treatment and prevention of neurodegenerative disorders, focusing on Alzheimer's disease.”

  1. Under the introduction, line 42, I suggest paraphrasing to “The discovery of lithium can be traced back to the mid-18th century and credited to”

Answer: The revised manuscript has been amended in accordance with your suggestions (Page 2, line 57-58).

“The discovery of lithium can be traced back to the mid-18th century and is credit-ed to José Bonifácio de Andrada e Silva, a Brazilian statesman who played a key role in his country's struggle for independence from Portugal.”

  1. Under the introduction, line 47, the sentence “After, samples of these minerals were spread to acquaintances in Sweden and other countries by Svendenstjerna” is complete or redundant. Please revise.

Answer: Thank you for your evaluation. The paragraph was indeed redundant and has been removed from the revised manuscript.

  1. Under the introduction, line 55, I suggest paraphrasing to “for the treatment of gout but was also commonly believed”

Answer: We appreciate your suggestion. The sentence has been rewritten for clarity and informativeness and can be found in the revised manuscript on page 2, lines 80-82.

“The early medical use of the lithium began with its administration for the treatment of gout, and later expanded to include conditions related to the bladder, kidneys, and gallstones.”

  1. Under the introduction, lines 68-69, I suggest paraphrasing to “In this experiment, the authors found/demonstrated that the urine of human subjects diagnosed with mania was harmful to animals”

Answer: Based on your recommendations, we have revised the sentences to enhance clarity and provide more detailed information, and can be found in the revised manuscript on page 3, lines 136-137.

“Although the mode of death in the animals continued to be consistent, Cade's experiment indicated that the urine of human subjects diagnosed with mania was harmful to animals compared to urine from other individuals.”

  1. Under the introduction, lines 72-73, I suggest paraphrasing to “a high dose of this solution elicited/produced anxiolytic effects in guinea pigs”

Answer: The sentence has been updated based on your suggestion, and it can be found on page 4, lines 151-152 of the revised manuscript.

“Cade also observed that when lithium salts were tested alone (i.e., without urea), they caused anxiolytic effects in guinea pigs.”

  1. Under the introduction, line 77, it will have a greater impact if the authors can briefly include the fundamentals that led to the advancement of this hypothesis.

Answer: We appreciate you bringing this to our attention. In response, we have included a brief overview of the key events that contributed to the development of Cade's hypothesis and his major findings. We believe the additional information will be of significant value to readers. The text added to the revised manuscript can be found on pages 2-4, highlighted in blue. Additionally, subtitles have been included to better guide the reader.

  1. Under the introduction, line 103, I suggest replacing “harmful effects’ with “unwanted effects”

Answer: We made the requested change (Page 5, line 208).

  1. Under the introduction, lines 111-112. A citation is necessary for this sentence. Also, the authors should refer to the publication by Welsseloo et al., British J. Psychiatry, 2017, titled “Lithium dosing strategies during pregnancy and postpartum period.”

Answer: Thank you for the valuable reference. We have included it in the revised manuscript, and it can be found on Page 5, lines 233-245.

“The control of lithium levels is particularly challenging during pregnancy due to normal physiological changes, especially in renal function. Increased glomerular filtration rate during pregnancy leads to decreased blood lithium concentrations and an in-creased risk of relapse in women with mania. As a result, clinicians often increase the lithium dose during pregnancy. However, later in pregnancy and during the early postpartum period, when glomerular filtration rate returns to preconception levels, the increased dosage may lead to toxic blood lithium levels [28]. Wesseloo et al. showed that lithium initiated during pregnancy, either as treatment for an episode or as an alternative to a mood stabilizer, reduced the risk of teratogenicity when started before conception and continued throughout pregnancy and the postpartum period. The lowest blood lithium levels occurred during the second trimester; in the third trimester and postpartum period, lithium levels progressively returned to preconception levels, highlighting the importance of controlling lithemic during this critical gestational period [28].”

  1. Under the introduction, line 118, what is natural lithium?

Answer: Thank you for your feedback. We have rewritten the text to clarify the concept, and it can be found on page 5, line 211-212 of the revised manuscript.

“Natural lithium is present in human nutrition through food intake, predominantly in plant-based foods and drinking water.”  

  1. Under the introduction, line 130, higher doses is more appropriate.

Answer: We have restructured the text in accordance with your suggestion (Page 5, lines 253-254). Thank you for your input.

Presumably, the adverse effects of lithium on cognition are associated with the use of higher doses [32–34].”

  1. Under the introduction, line 139, I suggest substituting “produce complete normality” with “restores normality”

Answer: We have made the suggested change, which can be found on pages 6, line 263 of the revised manuscript.

“But the interest in lithium's therapeutic action grew stronger when Paul Grof (1999) introduced the concept of 'excellent lithium responders', a specific subgroup of BD patients in whom lithium monotherapy can restore normality in clinical profile and cognitive function, compared to age-matched, healthy control subjects [36].”

  1. Under translational research, line 145, I will suggest deleting biological from biological mechanisms.

Answer: We made the suggested change (Page 6, lines 168-169).

“Despite its extensive clinical use over eight decades, uncertainties remain regarding the mechanisms underlying lithium's effects on mood and behavior.”

  1. Under translational research, line 167, I suggest paraphrasing to, “This indirect mechanisms activate”

Answer: We have made the suggested change, which can be found on pages 6, lines 291-292 of the revised manuscript.

“This indirect mechanisms activate intracellular kinases, such as Akt, while inhibiting the intracellular protein phosphatase 2A (PP2A) [46].”

  1. Under translational research, lines 185-186, I suggest paraphrasing to, “However, the exact mechanisms through which lithium modulates these factors are poorly understood” because multiple mechanisms of action may be at play in these processes.

Answer: We have made the recommended change, which can be found on page 6, lines 309-310.

“However, the exact mechanisms through which lithium modulates these factors are poorly understood.”

  1. Under lithium and neuroprotection, line 191, I suggest paraphrasing to “these patients show decreased prevalence of dementia compared to patients treated with other mood stabilizers”

Answer: The suggested change has been made, and it can be found on page 7, lines 315-316.

“Furthermore, these patients show decreased prevalence of dementia compared to patients treated with other mood stabilizers.”

  1. Under lithium and neuroprotection, line 192, I suggest paraphrasing to “suggesting that it may be due to lithium’s effects on brain systems responsible for maintaining homeostasis”

Answer: The recommended change has been implemented and can be found on page 7, lines 316-319.

“This benefit appears to be independent of therapeutic response parameters, suggesting that it may be due to lithium’s effects on brain systems responsible for maintaining homeostasis including second messenger pathways, cellular communication, and cell viability [59].”

  1. Under lithium and neuroprotection, lines 195-197, I suggest paraphrasing to “Lithium-mediated inhibition of GSJ+k-3 is causally linked to its effects on key pathogenic pathways of AD, particularly the A cascade and hyperphosphorylated tau (p-tau). The accumulation of A and p-tau is responsible for the formation of amyloid plagues and neurofibrillary tangles (NFTs), which are the key pathological hallmarks of AD”

Answer: We have made the recommended change, which can be found on page 7, lines 320-326.

“Lithium-mediated inhibition of GSK-3β is causally linked to its effects on key pathogenic pathways of AD, particularly the Aβ cascade and hyperphosphorylated tau (p-tau). The accumulation of Aβ and p-tau is responsible for the formation of amyloid plaques and neurofibrillary tangles (NFTs), which are the key pathological hallmarks of AD.”

  1. Under lithium and neuroprotection, line 202, this is the first time TCF4 is appearing in the text, therefore, it should be spelled out.

Answer: Thank you for bringing that to our attention. The modification was made to include the full form of the acronym in the text (Page 7, lines 327-330).

“As previously mentioned, elevated levels of nuclear β-catenin, caused by GSK-3β inhibition, result in its interaction with transcription Factor 4 (TCF4). This interaction inhibits the transcription of the main enzyme related to APP cleavage at the β 1 site (BACE1), ultimately reduces Aβ production [49].”

  1. Under lithium and neuroprotection, lines 203-204, I suggest paraphrasing to “ultimately reduces A production. Additionally, lithium attenuates the activity of”

Answer: We made the suggested change, Page 7, lines 329 -330.

“Additionally, lithium attenuates the activity of γ-secretase and enhances the clearance of Aβ by improving the function of brain-blood barrier (BBB) microvessels, promoting Aβ efflux through cerebrospinal fluid (CSF) bulk flow [26,37,60].”

  1. Under lithium and neuroprotection, for flow of ideas and continuity, lines 206-221 should be one paragraph. Also, in line 211, substitute pathology with neuropathology.

Answer: We have made the suggested changes throughout the manuscript (Page 7, lines 333-345).

  1. I suggest paraphrasing the beginning of the summary to “In summary, the neuroprotective actions of lithium against neuropathogenic mechanisms of AD and other neurodegenerative disorders are partly due to its intrinsic effects on oxidative stress, autophagy, apoptosis, mitochondrial function, and neuroinflammation.” Evidence in support of this hypothesis have emerged from the outcomes of translational”

Answer: We have made the recommended change, which can be found on page 7, lines 346-350.

“In summary, the neuroprotective actions of lithium against neuropathogenic mechanisms of AD and other neurodegenerative disorders are partly due to its intrinsic effects on oxidative stress, autophagy, apoptosis, mitochondrial function, and neuroinflammation. Evidence supporting this hypothesis has emerged from the outcomes of translational research [59].”

  1. On line 227, I suggest paraphrasing to “reducing transition to dementia in a population”

Answer: We made the suggested change (Page7, line 350-352).

“Controlled studies demonstrated that prolonged use of lithium carbonate in subtherapeutic doses could slow cognitive and functional decline while reducing transition to dementia in a population.”

  1. On line 230, I suggest paraphrasing to “Lithium is being explored as a potential disease modifying agent/treatment not only for BD and AD, but also for other neurodegenerative disorders such as…”

Answer: We have made the recommended change, which can be found on page 7, lines 354-355.

“Lithium is being explored as a potential disease modifying agent/treatment not only for BD and AD [63,65,66], but also for other neurodegenerative disorders such as Parkinson's disease [37], Huntington's disease [67], amyotrophic lateral sclerosis [68], and stroke [69].”

  1. On lines 234-235, I suggest paraphrasing to “and tissue concentration at the site of action. Consequently, some biological effects may occur at lower tissue concentrations than those typically encountered in the treatment of mood disorders.

Answer: We made the suggested change (Page7, line 358-360).

“The regulatory effects of lithium can vary based on factors such as the target tissue, duration of exposure, and tissue concentration at the site of action. Consequently, some biological effects may occur at lower tissue concentrations than those typically encountered in the treatment of mood disorders [33].”

  1. On line 241, I suggest paraphrasing to “suggest that the intake of lithium in low doses may result in long-term beneficial outcomes”

Answer: We made the suggested changes throughout the text (Page 8, lines 364-365).

“These findings suggest that the intake of lithium in low doses may result in long-term beneficial outcomes [75–79].”

  1. On line 245, I suggest paraphrasing to “These beneficial effects may be due to its effects on various cellular…”.

Answer: We made the suggested change throughout the text (Page 8, lines 369-371).

“These beneficial effects may be due to its effects on various cellular metabolic processes crucial for neuronal survival, neuroplasticity, transcriptional regulation, energy metabolism, and protection against neurotoxic damage.”

  1. On line 250, I suggest paraphrasing to “enhance intrinsic neuronal resilience and foster neuroprotective effects”

Answer: We made the suggested change throughout the text (Page 8, 373-374).

“Some of these mechanisms are directly associated with the central pathogenic processes of specific diseases, while others may reflect general responses that enhance intrinsic neuronal resilience and foster neuroprotective effects.”

  1. On line 251, I suggest deleted “currently.”

Answer: Thank you for bringing this to our attention. We have removed the word from the revised manuscript. 

Reviewer 2 Report

Comments and Suggestions for Authors

In this Mini-review, authors provide a detailed literature scan on lithium knowledge. A full and comprehensive introduction on historical basis has been provided.

Here, some advice to improve this article:

1- It would be easier to focus on the "type of the review" if it were reported in the title.

2- "Lithium is being explored as a potentially beneficial treatment not only for BD and AD, but also for other neurological conditions, such as Parkinson’s disease, Huntington's disease, amyotrophic lateral sclerosis, and stroke." paragraph 230-232. As listed in previous paragraphs, It would be helpful to better understand these trials. defining their outcomes, comparing their results in order to underline the key role of lithium, as stated in the objective of the present work.

3- some future direction would be intriguing to enrich scientific curiosity on such an ancient topic

Author Response

Responses to Reviewer #2

In this Mini-review, authors provide a detailed literature scan on lithium knowledge. A full and comprehensive introduction on historical basis has been provided.

Here, some advice to improve this article:

  1. It would be easier to focus on the "type of the review" if it were reported in the title.

Answer: Thanks for the suggestion, we changed the title to: 

“An Overview of the Effects of Lithium on Neurodegenerative Disorders: A Historical Perspective.”

  1. "Lithium is being explored as a potentially beneficial treatment not only for BD and AD, but also for other neurological conditions, such as Parkinson’s disease, Huntington's disease, amyotrophic lateral sclerosis, and stroke." paragraph 230-232. As listed in previous paragraphs, It would be helpful to better understand these trials. defining their outcomes, comparing their results in order to underline the key role of lithium, as stated in the objective of the present work.

Answer:  Thanks for the suggestion. However, we prioritize offering a more in-depth understanding of the historical perspective of lithium, which can be found on pages 2-4 of the revised manuscript. Expanding further on this topic would exceed the word limit set by the journal.

  1. Some future direction would be intriguing to enrich scientific curiosity on such an ancient topic

Answer: We appreciate your suggestion. A paragraph addressing this has been added to the revised manuscript and can be found on page 8, lines 380-386.

“In light of this, it is evident that future research should aim to further explore the biochemical mechanisms of lithium. Investigating the molecular pathways it influences will give us insight into its clinical efficacy and interaction with cellular processes. This enhanced knowledge will improve lithium's current use in mood disorders and possibly reveal new therapeutic targets. A more profound understanding of lithium's mechanisms will help develop more precise treatment strategies and novel pharmacological approaches.”

Reviewer 3 Report

Comments and Suggestions for Authors

The review "The effect of lithium in neurodegenerative disorders from a historical perspective" describes the role of lithium in neurodegenerative disorders and its role as a therapeutic agent. The review is compiled scientifically with suitable citations and explanations. However, some minor corrections are to be made before final publication. 

  1. What about the toxicity of lithium in human organs? is there no toxicity? Discuss effective doses and toxic doses from the literature for reference.
  2. Discuss the role of lithium in neurodegenerative disorders graphically. It will help the readers better understand the text. 
  3. Provide an author's perspective or future directions. Authors can include a table for clinical studies of lithium (Ongoing) in various neurological disorders, if any. 
  4. The role of lithium in different neurological disorders can be categorized under subheads for further clarification. 
  5. The manuscript needs further expansion with a detailed discussion of lithium in each discussed neurodegenerative condition. 
  6. The manuscript needs to be revised for minor grammatical and typo errors. 

Author Response

Responses to Reviewer #3

The review "The effect of lithium in neurodegenerative disorders from a historical perspective" describes the role of lithium in neurodegenerative disorders and its role as a therapeutic agent. The review is compiled scientifically with suitable citations and explanations. However, some minor corrections are to be made before final publication. 

  1. What about the toxicity of lithium in human organs? is there no toxicity? Discuss effective doses and toxic doses from the literature for reference.

Answer: We have made the recommended change, which can be found on page 8.

“Lithium was the first specific psychotropic agent, with sedatives such as bromides and paraldehyde, being its predecessors. Currently, it remains the benchmark for the treatment of bipolar disorder and is considered the standard reference among mood stabilizers. Although there are other approved medications for this condition, the use of lithium must be carefully assessed in comparison to alternative treatments, given that these alternatives may have differing safety and tolerability profiles. Nonetheless, the management of lithium treatment is challenging due to its narrow therapeutic index and the impact on kidney function, both of which heighten the risk of toxicity. Meticulous attention to dosing, monitoring, and titration is essential. Lithium is prescribed for managing acute manic and mixed episodes, as well as for long-term maintenance treatment in patients aged 12 or older. The therapeutic dose typically ranges from 300-2700 mg/day, with the optimal steady-state serum concentration of lithium falling between 0.7-1.2 mEq/L. In elderly individuals, lower doses of lithium are required, with a mean dose just above 400 mg/day, to achieve the desired serum concentration of approximately 0.5 mmol/L. For children over the age of 12, serum lithium concentrations should generally align with those observed in younger adults, ranging between 0.6-1.2 mmol/L. Serum concentrations of lithium exceeding 1.5 mEq/L may lead to symptoms of mild intoxication, including fatigue, tremor, nausea, diarrhea, blurred vision, vertigo, confusion, and increased deep tendon reflexes. Levels surpassing 2.5 mEq/L may cause more severe neurological complications, such as seizures, coma, cardiac dysrhythmia, and permanent neurological impairment, often affecting the cerebellum. Careful management is essential when using lithium, and it should be avoided in patients with a history of acute myocardial infarction, acute kidney failure, or certain rare heart rhythm disorders.”

  1. Discuss the role of lithium in neurodegenerative disorders graphically. It will help the readers better understand the text. 

Answer: Thank you for the suggestion. We developed a graphical abstract highlighting the key historical milestones of lithium and its principal neuroprotective effects. The graphical abstract was uploaded along with the other files.

  1. Provide an author's perspective or future directions. Authors can include a table for clinical studies of lithium (Ongoing) in various neurological disorders, if any. 

Answer: Thank you for the suggestion. As mentioned, we developed a graphical abstract highlighting the key historical milestones of lithium and its principal neuroprotective effects. The graphical abstract was uploaded along with the other files.

  1. The role of lithium in different neurological disorders can be categorized under subheads for further clarification. 

Answer: Thank you for your comment. The focus was primarily on Alzheimer's disease; however, we have also mentioned other neurodegenerative diseases and mental disorders. Page 10, lines 458-461.

“Lithium is being explored as a potential disease modifying agent/treatment not only for BD and AD, but also for other neurodegenerative disorders such as Parkinson's disease, Huntington's disease, amyotrophic lateral sclerosis, and stroke ”

  1. The manuscript needs further expansion with a detailed discussion of lithium in each discussed neurodegenerative condition. 

Answer: Thank you for your comment. The focus was primarily on Alzheimer's disease; however, we have also mentioned other neurodegenerative diseases and mental disorders. Page 10, lines 458-461. To clarify the study's objective, we revised the manuscript title by replacing “neurodegenerative diseases” with "Alzheimer's disease".

“Lithium is being explored as a potential disease modifying agent/treatment not only for BD and AD, but also for other neurodegenerative disorders such as Parkinson's disease, Huntington's disease, amyotrophic lateral sclerosis, and stroke ”

  1. The manuscript needs to be revised for minor grammatical and typo errors.

Answer: Thank you for bringing that to our attention. A comprehensive review of the English language has been conducted in the revised manuscript.

Reviewer 4 Report

Comments and Suggestions for Authors

Review of the Manuscript entitled:

 The effect of lithium in neurodegenerative disorders from a historical perspective

Dear Editor,

The authors in this review justified the repurposing of lithium as a candidate drug for the treatment of neurological conditions related to neurodegenerative processes and lesions affecting the central nervous system. The authors discussed the historical aspects of the discovery and early clinical experimentation with lithium in psychiatry. They reviewed its mechanisms of action and discussed the potential for treating and preventing neurodegenerative disorders, with emphasis on the modification of Alzheimer's disease (AD)…I think this article can be acceptable in Pharmaceuticals, but it needs some revisions for improving the quality of the manuscript:

  1. In the abstract, the authors only mentioned Alzheimer's disease. Did the authors encounter other mental disease which are affected by lithium ion?
  2. Please clarify the importance of lithium than other alkali metals such as sodium, potassium, rubidium or cesium and hybrid of them with lithium in the biofuel cells.
  3. Page 3, line 148: “Additionally, it affects the actions of GABA and NMDA receptors and contributes to calcium homeostasis [40,41].”

Please define the abbreviation of GABA and NMDA. Moreover, re-check for definition of other abbreviations in the whole manuscript.

  1. Page 3, lines 107-109: “Lithium is present in human nutrition based on food intake and is found predominantly in plant-derived foods and drinking water. Traces of lithium were detected in the human body in an average amount of approximately 7mg.”

Which kind of food do you mean? Please describe more.

  1. Page 4, line 163: what is GSK-3β? Please illustrate it.
  2. Why didn’t the authors investigate the energy of the biological systems with lithium? Stability energy and other extract parameters can validate the work more.
  3. Did the authors encounter the existence and function of lithium in cell membrane? Please clarify it.
  4. Before the conclusion, please consider a short paragraph as the consequence of all sections to indicate the final discussion of your work.
  5. It is recommended that the authors discuss electron density, electron transfer and the conduction potential of lithium in biological cells.
  6. Please categorize and improve the conclusion section with more significant keys.
  7. Please re-read the whole manuscript to correct grammatical mistakes, typos or any mistakes.

Author Response

Responses to Reviewer #4

The authors in this review justified the repurposing of lithium as a candidate drug for the treatment of neurological conditions related to neurodegenerative processes and lesions affecting the central nervous system. The authors discussed the historical aspects of the discovery and early clinical experimentation with lithium in psychiatry. They reviewed its mechanisms of action and discussed the potential for treating and preventing neurodegenerative disorders, with emphasis on the modification of Alzheimer's disease (AD)…I think this article can be acceptable in Pharmaceuticals, but it needs some revisions for improving the quality of the manuscript:

  1. In the abstract, the authors only mentioned Alzheimer's disease. Did the authors encounter other mental disease which are affected by lithium ion?

Answer: Thank you for your comment. The focus was primarily on Alzheimer's disease; however, we have also mentioned other neurodegenerative diseases and mental disorders. Page 10, lines 458-461. To clarify the study's objective, we revised the manuscript title by replacing “neurodegenerative diseases” with "Alzheimer's disease".

“Lithium is being explored as a potential disease modifying agent/treatment not only for BD and AD, but also for other neurodegenerative disorders such as Parkinson's disease, Huntington's disease, amyotrophic lateral sclerosis, and stroke ”

  1. Please clarify the importance of lithium than other alkali metals such as sodium, potassium, rubidium or cesium and hybrid of them with lithium in the biofuel cells.

Answer:  Thanks for the suggestion. We made the suggested change throughout the text (Pages 6-7, 294-317).

“Lithium is an essential trace element involved in various cellular processes, influencing the functions of multiple enzymes, hormones, and vitamins. The fact that lithium is a monovalent cation with chemical properties similar to potassium (K⁺) and Na⁺, suggests that it plays a key role in maintaining brain homeostasis. Lithium is equally distributed between intra- and extracellular compartments and enters cells mainly through voltage-gated Na⁺ channels, with its influx exceeding its efflux. Since lithium enters cells through electrical signaling, it accumulates more selectively in neurons with higher synaptic activity. As a result, lithium is more likely to interact with neurotransmitters and receptors in the brain, modulating neuronal plasticity, membrane components, transport mechanisms, and intracellular signaling molecules. At a structural level, it is via such a mechanism that lithium appears to preserve or increase the volume of brain structures involved in emotional regulation such as the prefrontal cortex, hippocampus and amygdala, possibly reflecting its neuroprotective effects.”  …  “In addition to lithium, the others alkali metals, including Na⁺, K⁺, rubidium (Rb), cesium (Cs), and francium (Fr), have some function in the organism. In the human body, only Na+ and K+ are essential and vital to life, playing crucial roles in various physiological processes, such as the conduction of electrical impulses in excitable cells. Even a slight variation in their concentrations, either in the intra- or extracellular compartments, can lead to acute dysfunction of vital organs such as the brain, heart, and kidneys, potentially resulting in a life-threatening state. In contrast, Rb and Cs are considered non-essential elements for humans and have no known biological role.”

  1. Page 3, line 148: “Additionally, it affects the actions of GABA and NMDA receptors and contributes to calcium homeostasis [40,41].” Please define the abbreviation of GABA and NMDA. Moreover, re-check for definition of other abbreviations in the whole manuscript.

Answer:  We have made the recommended change, which can be found on page 8, lines 377-378.

“Additionally, it affects the actions of GABA (gamma-aminobutyric acid) and NMDA (N-methyl-D-aspartate) receptors, and contributes to calcium homeostasis.”

  1. Page 3, lines 107-109: “Lithium is present in human nutrition based on food intake and is found predominantly in plant-derived foods and drinking water. Traces of lithium were detected in the human body in an average amount of approximately 7mg.” Which kind of food do you mean? Please describe more.

Answer: Thank you for your comment. We have made the suggested changes in the revised manuscript, which can be found on page 7, lines 319-323.

“Low-level exposure to naturally occurring lithium in the environment, such as in drinking water and certain food, including vegetables, fruits, and grains—e.g., cauli-flower, tea plants, coriander leaves, tomatoes, garlic, nutmeg, cumin seeds, onions, green chilies, rice, and wheat—has been shown to benefit mental health.”

  1. Page 4, line 163: what is GSK-3β? Please illustrate it.

Answer: Thank you for bringing that to our attention. The modification was made to include the full form of the acronym in the text (Page 8, line 381).

The primary molecular targets of lithium are the enzymes inositol monophosphatase (IMPase) and glycogen synthase kinase-3 beta (GSK-3β).”

  1. Why didn’t the authors investigate the energy of the biological systems with lithium? Stability energy and other extract parameters can validate the work more.

Answer: Thank you for the suggestion. We have added more information on the topic in the revised manuscript (Page 7, lines 329-335)

“Another important characteristic of lithium is its ability to easily distribute throughout the body via both intra- and extracellular fluids, due to its free protein binding. Lithium can cross the placenta to reach the developing fetus, and is also secreted in breast milk. As a result, lithium concentrations in both mothers and fetuses attain equilibrium, suggests that it may contribute to neurodevelopmental health. In addition, during breastfeeding, the lithium concentration in milk decreases to ap-proximately half of the maternal serum levels.”

  1. Did the authors encounter the existence and function of lithium in cell membrane? Please clarify it.

Answer: Thank you for your question. We have made the suggested changes in the revised manuscript, which can be found on pages 6-7, lines 294-307.

“Lithium is an essential trace element involved in various cellular processes, influencing the functions of multiple enzymes, hormones, and vitamins. The fact that lithium is a monovalent cation with chemical properties similar to potassium (K⁺) and Na⁺, suggests that it plays a key role in maintaining brain homeostasis. Lithium is equally distributed between intra- and extracellular compartments and enters cells mainly through voltage-gated Na⁺ channels, with its influx exceeding its efflux. Since lithium enters cells through electrical signaling, it accumulates more selectively in neurons with higher synaptic activity. As a result, lithium is more likely to interact with neurotransmitters and receptors in the brain, modulating neuronal plasticity, membrane components, transport mechanisms, and intracellular signaling molecules. At a structural level, it is via such a mechanism that lithium appears to preserve or increase the volume of brain structures involved in emotional regulation such as the prefrontal cortex, hippocampus and amygdala, possibly reflecting its neuroprotective effects.”

  1. Before the conclusion, please consider a short paragraph as the consequence of all sections to indicate the final discussion of your work.

Answer: Thank you for bringing that to our attention. A paragraph addressing this has been added to the revised manuscript and can be found on page 10.

“The evolution of lithium therapy highlights the progression of psychiatric medicine—from initial skepticism to its establishment as an essential therapeutic tool—underscoring the importance of continued research and clinical advancements in mental health treatment.”

  1. It is recommended that the authors discuss electron density, electron transfer and the conduction potential of lithium in biological cells.

Answer: Thank you for the suggestion, new information has been added throughout the text and was and was referenced in the question above.

  1. Please categorize and improve the conclusion section with more significant keys.

Answer: We have made the recommended change, which can be found on page 10.

“It is important to note that the clinical use of lithium should be limited to conditions where its efficacy has been well-established by scientific evidence, such as mood disorders. For other health conditions, further research is needed to support new therapeutic indications. Lithium treatments, including off-label use, should consider factors such as dosage, the specific health condition, age, and potential drug interactions.”

  1. Please re-read the whole manuscript to correct grammatical mistakes, typos or any mistakes.

Answer:  Thank you for the suggestion, the text was reread and improved.